# The relationship between environmental sources and the susceptibility of *Acanthamoeba* keratitis in the United Kingdom

Nicole A. Carnt[1,2,3]*, Dinesh Subedi [1,4], Sophie Connor[5], Simon Kilvington[6]

**1** School of Optometry and Vision Science, University of New South Wales, Sydney, Australia, **2** Westmead Institute for Medical Research, University of Sydney, Sydney, Australia, **3** University College London Institute of Ophthalmology, London, England, United Kingdom, **4** School of Biological Sciences, Monash University, Clayton, Australia, **5** Research Organisation (KC) Ltd, London, England, United Kingdom, **6** Ophtecs Corporation, Kobi, Japan

* n.carnt@unsw.edu.au

## Abstract

### Purpose

To determine whether *Acanthamoeba* keratitis (AK) patients have higher rates of *Acanthamoeba* and free-living amoeba (FLA) colonising domestic sinks than control contact lens (CL) wearers, and whether these isolates are genetically similar to the corneal isolates from their CL associated AK.

### Methods

129 AK patients from Moorefield Eye Hospital, London and 64 control CL wearers from the Institute of Optometry were included in this study. The participants self-collected home kitchen and bathroom samples from tap-spouts, overflows and drains using an instructional kit. The samples were cultured by inoculating onto a non-nutrient agar plate seeded with *Escherichia coli*, incubated at 32˚C and examined for amoebae by microscopy for up to 2 weeks. Partial sequences of mitochondrial cytochrome oxidase genes (*coxA*) of *Acanthamoeba* isolates from four AK patients were compared to *Acanthamoeba* isolated from the patient's home. The association between sampling sites was analysed with the chi-square test.

### Results

A total of 513 samples from AK patients and 189 from CL controls were collected. The yield of FLA was significantly greater in patients' bathrooms (72.1%) than CL controls' bathrooms (53.4%) (p<0.05). Spouts (kitchen 6.7%, bathroom 11%) had the lowest rate of *Acanthamoeba* isolation compared to drains (kitchen 18.2%, bathroom 27.9%) and overflow (kitchen 39.1%, bathroom 25.9%) either in kitchens or bathrooms (p<0.05). There was no statistically significant difference between the average prevalence of *Acanthamoeba* in all three sample sites in kitchens (16.9%) compared to all three sample sites in bathrooms (21.5%) and no association for *Acanthamoeba* prevalence between AK patients and CL controls. All four

**Data Availability Statement:** All relevant data are within the manuscript and its Supporting Information files.

**Funding:** Dr Carnt and Dr Kilvington received Dallos Award 2015 from British Contact Lens Association (BCLA) to conduct this study. Dr Carnt was supported by an NHMRC Early Career CJ Martin Fellowship (APP1036728), National Institute of Research (NIHR) Biomedical Research Centre (BRC) at Moorfields Eye Hospital NIHR BRC. The views expressed are those of the author(s) and not necessarily those of the BCLA, the NHS, the NIHR or the Department of Health. The funders provided support in the form of salaries for author NC and research materials but did not have any additional role in the study design, data collection and analysis, decision to publish, or preparation of the manuscript.

**Competing interests:** SC is employed by Research Organisation (KC) Ltd. SK is employed by Ophtecs Corporation. There are no patents, products in development, or marketed products to declare. The commercial affiliation does not alter our adherence to PLOS ONE policies on sharing data and materials.

corneal isolates had the same *coxA* sequence as at least one domestic water isolate from the patients' sink of the kitchen and the bathroom.

## Conclusion

The prevalence of *Acanthamoeba* and FLA was high in UK homes. FLA colonisation was higher in AK patients compared to controls but the prevalence of *Acanthamoeba* between AK patients and CL controls domestic sinks was similar. This study confirms that domestic water isolates are probably the source of AK infection. Advice about avoiding water contact when using CL's should be mandatory.

## Introduction

Free-living amoebae (FLA) are unicellular eukaryotic organisms that can grow independently in different environments, including natural and man-made bodies of water; lakes, ponds, swimming pools, and even treated water supplies [1–3]. Some genera of FLA such as *Acantha-moeba*, *Vahlkampfia*, *Naegleria and Hartmannella* are opportunistically pathogenic to humans [1, 4].

The term *Acanthamoeba* keratitis (AK) refers to infection of the cornea by *Acanthamoeba*. However, other FLA such as *Vahlkampfia* and *Hartmannella* are also known causative agents of keratitis [4, 5]. AK and other amoebal keratitis are increasingly being recognized as a severe ocular infection worldwide that occurs most often among contact lens (CL) wearers and can lead to blindness [6–10]. Water contamination has been recognized as the most important risk factor for CL-associated AK [11–14].

*Acanthamoeba* keratitis was first reported in 1974 as an extremely rare disease [15]. With the increased population of CL wearers, the incidence of AK has significantly risen [16, 17]. The reported incidence of AK in developed countries is up to 149 cases per million per year for contact lens wearers but it is less than 2 per million per year for non-contact lens wearers [18, 19]. In an outbreak in England and Wales during 1997–1999, the annual incidence of AK was 1.13 (in general adults) to 21.14 per million (in CL wearers) [18]. The latter study also found that the incidence of AK was much greater in areas supplied with hard water, which enhances limescale formation on pipes and so increases colonisation of *Acanthamoeba* [18]. Furthermore, distance from water purification plants, use of stagnant water (for example cis-terns), and warmer air temperature were found can be associated with higher incidence of AK [14, 19–26]. In a more recent outbreak that started in the UK in 2010–2011 a three-fold increase in the incidence of AK was reported compared to the outbreak in 2004–2009 [27]. The increased number of AK cases in the UK has been linked to increased use of disposable contact lenses in case control studies [28, 29] and improper lens hygiene [29].

Kilvington *et al.* have found 89% of patients with culture-positive AK contained FLA including *Acanthamoeba* in tap water from their kitchens, or bathrooms, and water storage tanks were implicated as promoting this colonisation [11]. *Acanthamoeba* were cultured from 30% of all homes, and 75% of isolates from domestic water and isolates from the corneas of AK patients had identical mtDNA profiles [11]. However, that study did not examine water samples from CL wearers who were not AK patients. Such sampling may help the understand-ing of the CL-wearing population's risk of developing AK. In the current study, samples from both AK patients and control CL wearers and from different areas of their kitchen and bath-room sinks were cultured to understand whether AK patients have higher rates of

*Acanthamoeba* sink colonisation than control CL wearers. The current study also aimed to determine the differences in the prevalence of *Acanthamoeba* and FLA in spouts, overflows and drains of kitchens and bathrooms, whether *Acanthamoeba* colonisation of domestic water systems remained constant over time and whether domestic waterborne *Acanthamoeba* isolates were similar to those isolated from cases of AK.

# Methods

## Sample collection

A total of 129 AK patients from Moorfield's Eye Hospital, London and 64 control CL wearers from Institute of Optometry, London were included in this study. The research protocol received approval from the National Research Ethics Service Committee London-Hampstead (REC reference 13/LO/0032) and the Moorfields Eye Hospital Research governance committee. Written informed consent was received from participants before initiation of the study. Each participant was provided with a sampling pack containing six sterile polyester-tipped applicators and sterile screw-cap test tubes, written instructions (Supplementary information S1 Data) and a questionnaire which included questions on the suburb, date and time of sample collection, and the date and time that samples were returned to researchers. Participants were requested to swab the inside of their bathroom and kitchen spouts, sink drains and overflows for 10 seconds with the applicator, place the swab into the test tube, fill the tube with 5mL cold tap-water, then fasten the test tube cap tightly. A total of 23 repeat samples at least one month apart were collected from patient's kitchen and bathroom.

## Culture and microscopy

Upon receipt at the laboratory, tubes were vortexed and 500 μL of water was inoculated onto 1.4% non-nutrient agar (NNA) plate pre-seeded with 100μL of viable *Escherichia coli*. Each agar plate was then incubated at 32˚C in sealed polythene bags. After incubation for 3–4 days, plates were examined daily for up to 10 days using an inverted light microscope for the presence of FLA and *Acanthamoeba*. Isolates were identified by morphologic examination of the trophozoite and cyst forms [30]. Samples identified with *Acanthamoeba* were classified as *Acanthamoeba* positive and those with *Acanthamoeba* or FLA were classified as FLA in current analysis.

## PCR assay, *coxA* sequencing and sequence homology analysis

Nucleotide sequence of the mitochondrial cytochrome oxidase subunit-1 and -2 (*cox1/2*) of corneal isolates were compared with isolates from the patients' homes. DNA was extracted using Chelex resin (MB Chelex-100 resin; Bio-Rad Laboratories, Hercules, CA, USA) following the method described by Kilvington *et al*.[31]. *Cox1/2* was amplified by PCR using previously established primer and cycle conditions [11]. The amplified products were sent for Sanger sequencing and DNA sequences were aligned using ClustalW and a phylogenetic tree constructed using MEGA 7 [32].

## Statistical analysis

The Pearson Chi-square test was used to assess whether there was a statistically significant difference in the association between sampling sites. Odds ratios and their 95% confidence intervals (95% CIs) were calculated to measure association between AK cases and detection rate of *Acanthamoeba* and FLA in AK patients' and CL controls' homes.

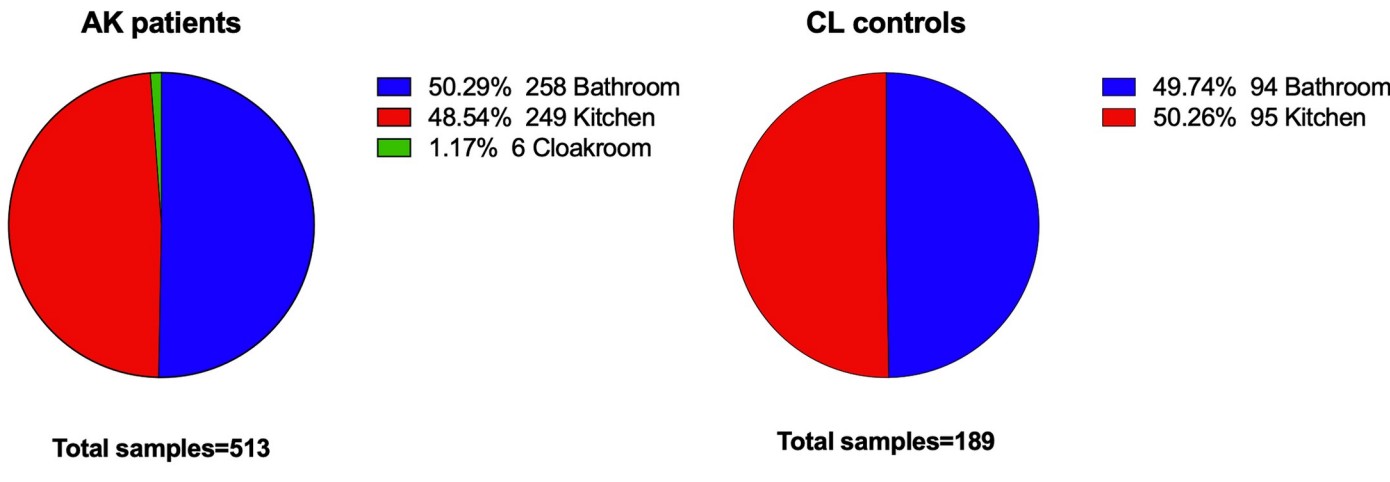

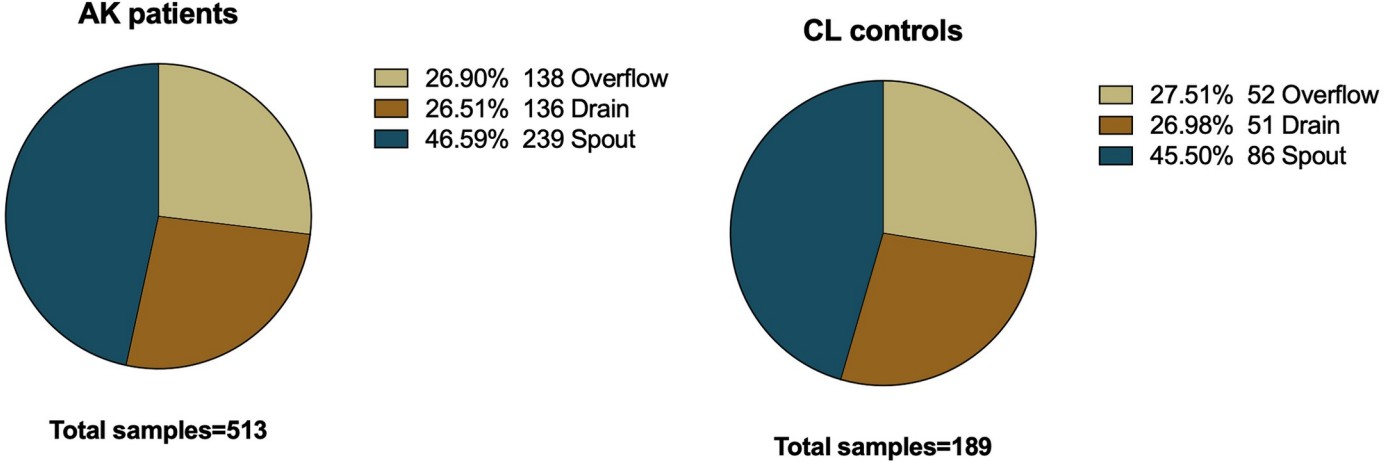

**Fig 1. Distribution of samples analysed in current study.**

## Results

### *Acanthamoeba* and free living amoeba colonisation

A total of 513 samples from 77 AK patients and 189 samples from 40 CL controls were retrieved and examined in this study. Samples were collected from water tap-spouts, sink overflows and drains from the kitchen and bathroom. The proportion of samples from kitchens and bathrooms were broadly similar (Fig 1). Samples collected from the cloakrooms of AK patients were excluded from the current analysis because the number of samples were small (1.7%) and there were no cloakrooms samples from CL controls group.

A slightly higher proportion of *Acanthamoeba* were cultured from bathrooms (average 21.5%) compared to kitchens (average16.9%) (Fig 2) and there was no difference in this proportion between the patient and control group. However, bathrooms yielded a higher proportion of FLA positive samples compared to kitchens. Bathrooms from the AK cohort had a

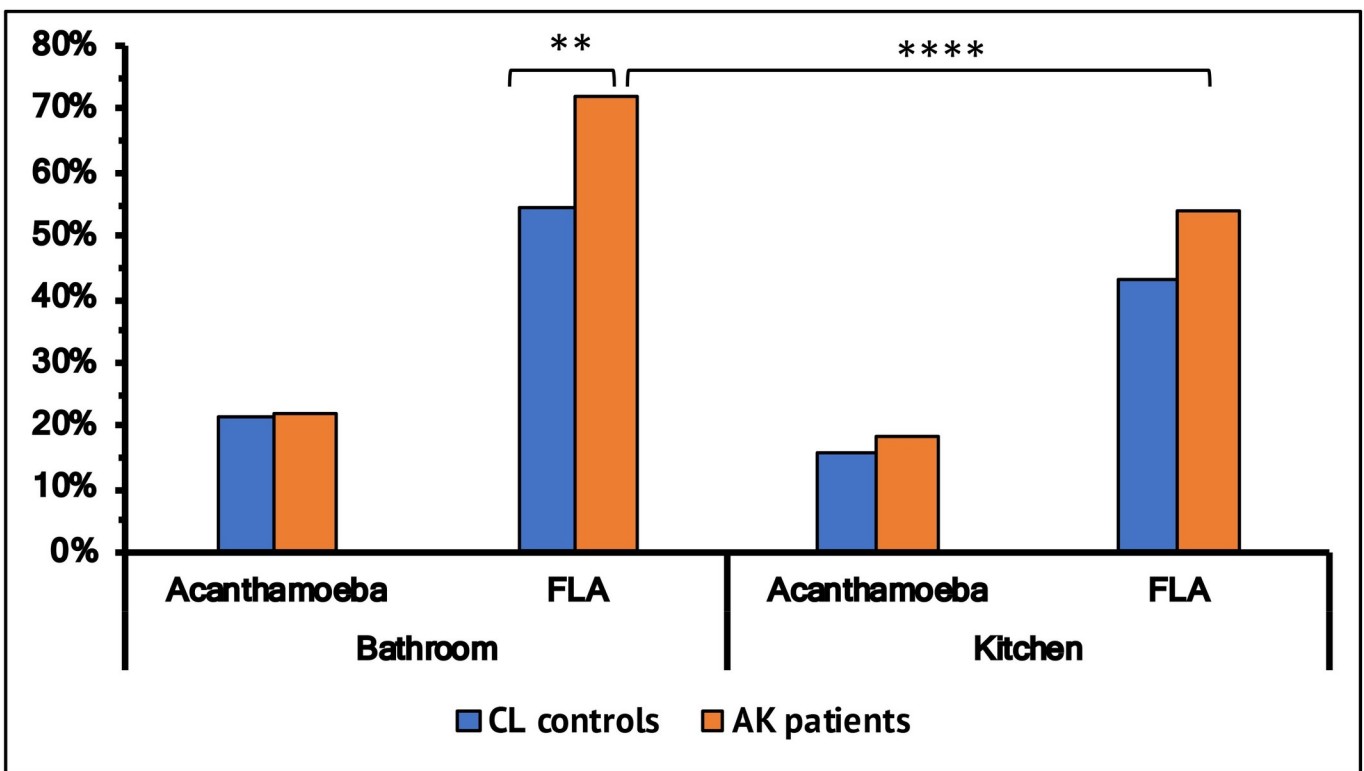

**Fig 2. Overall colonisation of *Acanthamoeba* and free-living amoeba (FLA) in samples from bathroom and kitchen in patients and control groups.** * denotes level of significance (*p ≤ 0.05, **p ≤ 0.01, ***p ≤ 0.001, ****p ≤ 0.0001).

higher proportion (72.1%) of FLA than samples from the control's bathrooms (54.3%) ($p < 0.05$) (OR 2.1; 95% CI 1.33–3.55). Furthermore, AK patient's bathrooms had a significantly higher proportion of FLA than kitchens ($p < 0.05$) (Fig 2). Controls had a similar higher proportion of FLA from bathrooms, but this did not reach statistical significance.

Colonisation of *Acanthamoeba* in tap-spouts was lower in both bathrooms and kitchens ($p < 0.05$) than drains or overflows, but there was no statistical difference between patients and controls in the prevalence of *Acanthamoeba* in these sites (Fig 3). On the other hand, the rate of colonisation of FLA was higher in drains, overflows and tap-spouts of patients' kitchens and bathrooms compared to these three sites from controls, and this reached statistical significance between bathroom's overflow (controls 14/27 vs patients, 55/72, $p = 0.02$) and kitchen's drain (controls 8/25 vs patients 40/66, $p = 0.02$) (Fig 4).

Twenty-three repeat samples at the two-sampling time-points for both bathrooms and kitchens spouts were examined to understand whether *Acanthamoeba* was consistently present at the same sites. *Acanthamoeba* isolation appeared to be sporadic from either bathrooms or kitchens (Table 1).

### Patients and environmental *Acanthamoeba* colonisation

Four AK patients were selected to the study genetic relatedness between pathogenic and environmental isolates on the basis of association between case and detection of *Acanthamoeba* in water samples from their homes. Partial sequence of the mitochondrial cytochrome oxidase gene (*coxA 1/2*) was obtained from *Acanthamoeba* isolates of four different keratitis patients and those sequences were compared with isolates from the patient's home. Fig 5 shows that

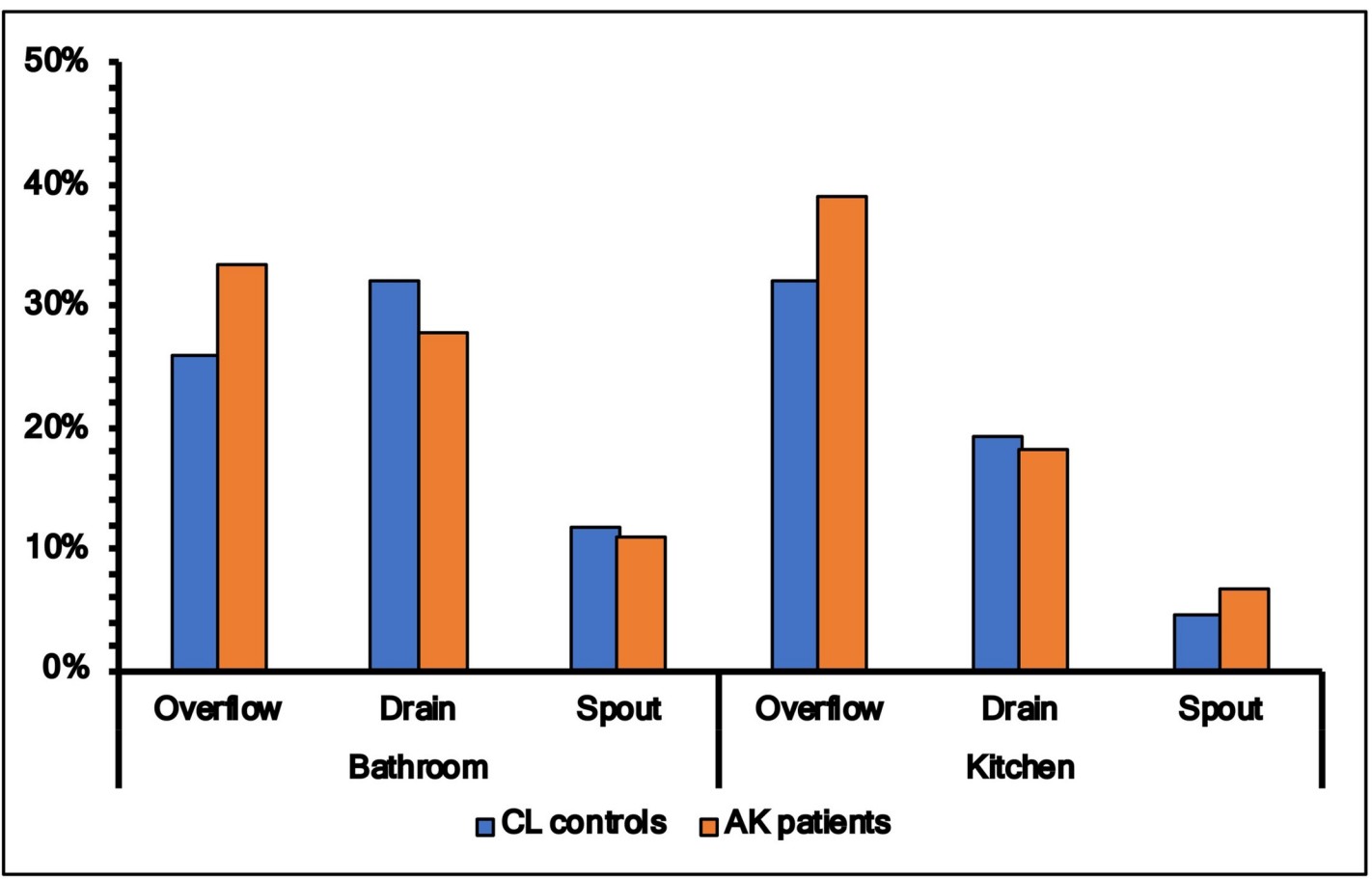

**Fig 3. Colonisation of *Acanthamoeba* in samples for spout, drain and overflow from bathroom and kitchen in patients and control groups.**

each pathogenic isolate had the identical *coxA 1/2* sequence to at least one domestic water isolate including isolates from both the kitchen and the bathroom. The *Acanthamoeba* isolate from patient ID78 was similar to isolates from the patient's bathroom spout, drain and overflow and their kitchen's overflow suggesting a possible source of keratitis isolate. This isolate was also similar to the corneal isolate from another patient ID47 and an isolate from that patient's kitchen spout, suggesting possibility of common source of infections. For patients ID81 and ID37, their corneal isolates were similar to those from their bathrooms. Water contact was reported by three of these four patients in the days prior to suffering AK.

## Discussion

Domestic tap water is a major risk factor for contact-lens associated *Acanthamoeba* keratitis [11, 18, 20] and is a reservoir for *Acanthamoeba* and other FLA [11]. Based on *coxA 1/2* gene sequence [33], this study identified that isolates from water sources were very similar to isolates from the corneas of AK patients, suggesting that domestic waterborne *Acanthamoeba* could be associated with AK. The current study examined water samples from CL wearers who were not AK patients and compared these to samples from AK patients. There was no variation in the rate of colonisation of *Acanthamoeba* in kitchens and bathrooms of AK patients compared to that of CL wearers controls, suggesting that all contact lens wearers are at risk of developing AK and it is not that AK patients have bathrooms or kitchens that are more

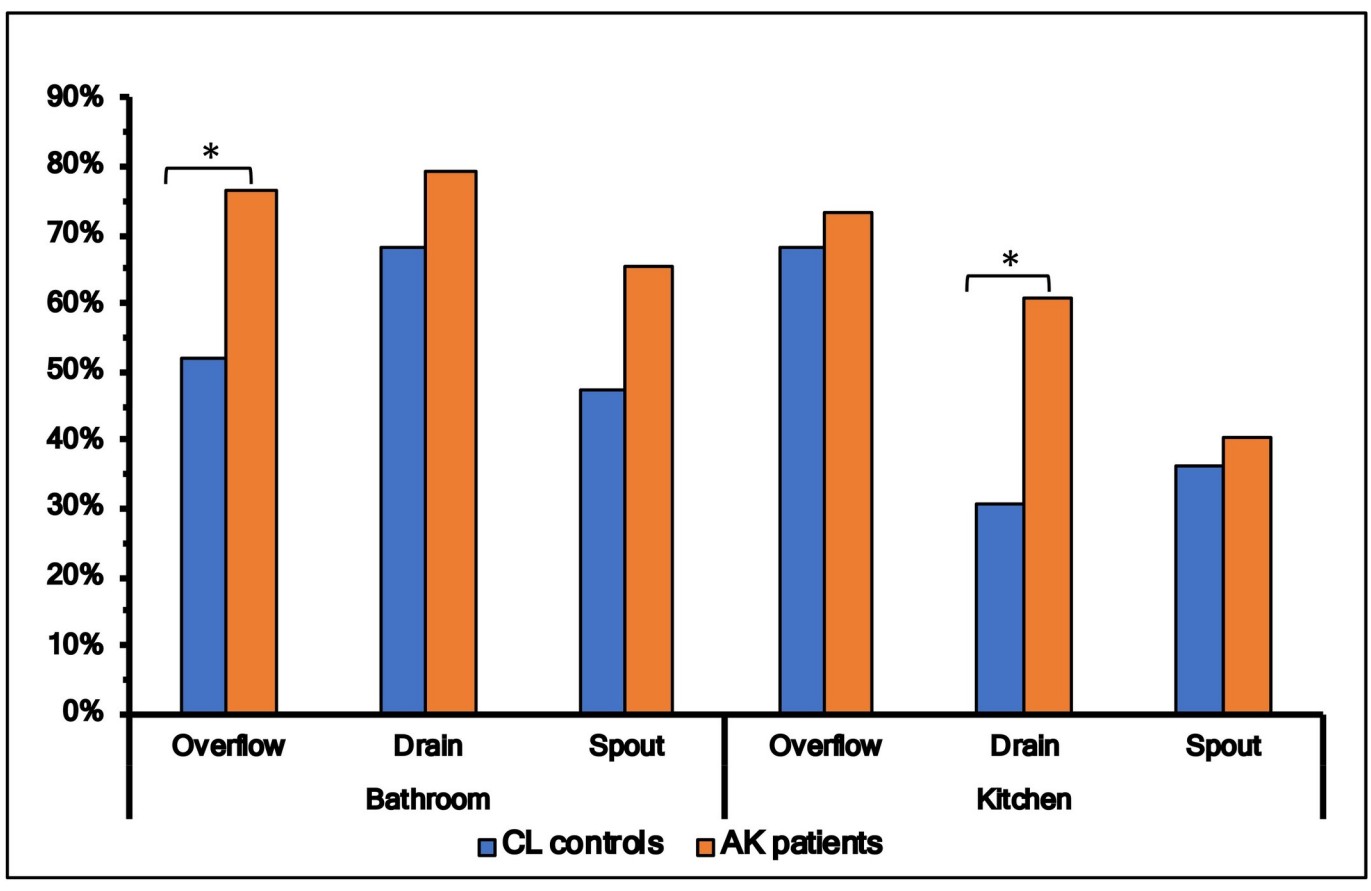

**Fig 4. Colonisation of free-living amoeba (FLA) in samples for spout, drain and overflow from bathroom and kitchen in patients and control groups.** * denotes level of significance (*p ≤ 0.05, **p ≤ 0.01, ***p ≤ 0.001, ****p≤ 0.0001).

frequently colonised. Patient's bathrooms yielded a higher proportion of FLA compared to the bathrooms of control CL wearers or the kitchens of patients. Poorly maintained sinks may allow microbial biofilms to form, and this is believed to facilitate amoebal colonisation of water outlets and drains [11], not least because amoebae can graze of the bacteria in biofilms [34].

The prevalence of *Acanthamoeba* observed in this study (21.5%) was slightly lower than the prevalence reported by another UK based study, in which Kilvington *et al*. isolated *Acanthamoeba* from 30% of homes [11]. In the current study tap spouts tended to have filters on them making it difficult to swab the inside of spouts and this could be the reason for these differences [11]. In addition, spouts have the fastest water flow and large volume of water regularly flows through it, which self-flushes the spouts and may prevent colonisation of FLA.

**Table 1. Repeatability of samples.**

| | Bathroom | | | | Kitchen | | |
|---|---|---|---|---|---|---|---|
| | | **First Sample** | | | | **First Sample** | |
| | | Positive | Negative | | | Positive | Negative |
| **Second samples** | Positive | 1 | 2 | **Second samples** | Positive | 0 | 3 |
| | Negative | 3 | 17 | | Negative | 3 | 17 |

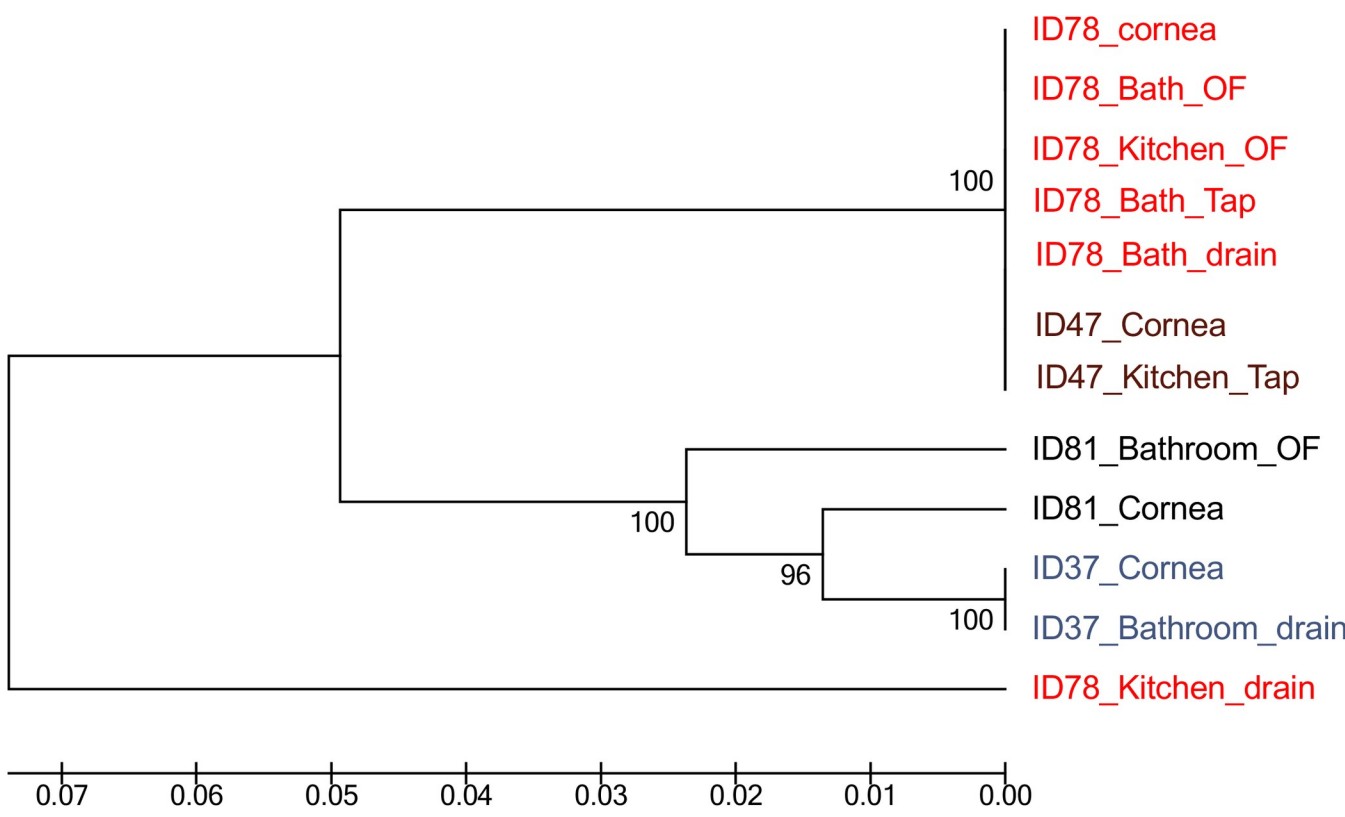

**Fig 5. Cytochrome oxidase gene (*coxA 1/2*) sequence comparison of patient and environmental *Acanthamoeba* isolates.**

Furthermore, there is a wide variation in the rate of isolation of *Acanthamoeba* in previous reports. Studies based in Hong Kong have shown that household tap water yielded 10% [35], 8.3% [36] and 7.7% [36] *Acanthamoeba* positive samples. In addition, studies from Scotland (12% bathrooms, 2% kitchen) [37], Jamaica (36%, tap-water) [38] and Florida (2.8%, domestic water) [39] have reported various isolation rates in water samples. One of the reasons for these variations may be due to use of different methods for identification of *Acanthamoeba*, for example morphological [39] versus molecular [38] techniques. Given that PCR-based methods are more sensitive than culture for detection of *Acanthamoeba* [40], further studies based on molecular identification will be required to validate rate of isolation of *Acanthamoeba* observed in the current study.

The domestic water supply system of the UK, which uses water storage cisterns as well as often having hard water, is believed to be a cause for the higher prevalence of *Acanthamoeba* in the tap water in this region [11]. *Acanthamoeba* and other microbes proliferate in the cistern water. Also, there is a seasonal trend with the colonisation of *Acanthamoeba* increasing in warmer months [22–24].

A limitation of this study is that the genus-level identification of FLA by molecular methods was not performed. FLA such as *Hartmannella* and *Vahlkampfiid* amoebae [4, 5], although rare compared to *Acanthamoeba*, are also causative agents of keratitis. In addition, the significantly higher FLA positive samples in the patient cohort suggests that their identification in the genus-level will help to better understand the risk of keratitis associated with domestic tap water. Further studies regarding the pathogenicity of FLA and *Acanthamoeba* will also require assessing the source of transmission as well as the severity of AK.

In conclusion, this study suggested that *Acanthamoeba* colonisation in UK water supply is high and occurs in CL-wearers and AK patients. Domestic water isolates were similar to isolates from the cornea of AK patient, confirming that domestic waterborne *Acanthamoeba* is still associated with keratitis. Advice about avoiding domestic water contact when using CL's should be mandatory.

## Supporting information

**S1 Data. Water sample instruction.**
(DOCX)

## Acknowledgments

The authors would like to thank Ms Varshini Parayoganathan for database management, Ms Melanie Mason and Mr Scott Hau, Dr Judith Morris and Institute of Ophthalmology for subject recruitment. We would also like to thank Prof John Dart, Moorefields Eye Hospital for providing guidance and recruitments and Prof Mark Willcox, School of Optometry and Vision Science, UNSW Sydney for editing the draft.

## Author Contributions

**Conceptualization:** Nicole A. Carnt, Simon Kilvington.

**Data curation:** Nicole A. Carnt, Dinesh Subedi, Sophie Connor.

**Formal analysis:** Nicole A. Carnt, Dinesh Subedi, Simon Kilvington.

**Funding acquisition:** Nicole A. Carnt, Simon Kilvington.

**Investigation:** Nicole A. Carnt, Sophie Connor.

**Methodology:** Nicole A. Carnt, Simon Kilvington.

**Project administration:** Nicole A. Carnt, Simon Kilvington.

**Resources:** Simon Kilvington.

**Supervision:** Simon Kilvington.

**Visualization:** Simon Kilvington.

**Writing – original draft:** Nicole A. Carnt, Dinesh Subedi.

**Writing – review & editing:** Nicole A. Carnt, Dinesh Subedi.

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
