## [Decision Letter · Decision Letter 0]

31 Dec 2019

PONE-D-19-26513

The relationship between environmental sources and the susceptibility of Acanthamoeba keratitis in the United Kingdom

PLOS ONE

Dear Dr Carnt,

Thank you for submitting your manuscript to PLOS ONE. After careful consideration, we feel that it has merit but does not fully meet PLOS ONE’s publication criteria as it currently stands. Therefore, we invite you to submit a revised version of the manuscript that addresses the points raised during the review process.

We would appreciate receiving your revised manuscript by Feb 14 2020 11:59PM. To enhance the reproducibility of your results, we recommend that if applicable you deposit your laboratory protocols in protocols.io, where a protocol can be assigned its own identifier (DOI) such that it can be cited independently in the future. For instructions see: http://journals.plos.org/plosone/s/submission-guidelines#loc-laboratory-protocols

We look forward to receiving your revised manuscript.

Kind regards,

Manuel Garza León

Academic Editor

PLOS ONE

Journal Requirements:

'British Contact Lens Association, Dallos award, 2015 (Carnt and Kilvington). Dr

Carnt was supported by an NHMRC Early Career CJ Martin Fellowship (APP1036728).  The views expressed are those of the author(s) and not necessarily those of the NHS, the NIHR or the Department of Health'

We note that one or more of the authors are employed by a commercial company: 

Research Organisation (KC) Ltd, and Ophtecs Corporation.

Reviewers' comments:

Reviewer's Responses to Questions

**Comments to the Author**

1. Is the manuscript technically sound, and do the data support the conclusions?

Reviewer #1: Yes

Reviewer #2: Yes

2. Has the statistical analysis been performed appropriately and rigorously? 

Reviewer #1: Yes

Reviewer #2: No

3. Have the authors made all data underlying the findings in their manuscript fully available?

Reviewer #1: Yes

Reviewer #2: Yes

4. Is the manuscript presented in an intelligible fashion and written in standard English?

Reviewer #1: Yes

Reviewer #2: Yes

5. Review Comments to the Author

Reviewer #1: Line 138-139 – Spouts have the fastest flowing and largest volume of water flowing through it with thousands of gallons per month, this may be the reason why you do not see as much bacteria/Acanthamoeba/FLA in that particular area for Fig 3. Should repeat samples be taken from the drain or overflows from these areas for future studies? Please comment/discuss this finding.

Do you think that paranoia of patients when sampling they didn’t want to see Acanthamoeba or FLA within their home that they may have stringently cleaned between sampling?

All figures/graphs are of poor quality please use another software program.

Were attempts to characterize the species/isolate causing the AK in these 4 patients?

Supplemental information is clear for samplers.

Molecular characterization of FLA should become mandatory in future studies.

Reviewer #2: Overall, it is a very well written case-control study on the association between environmental sources of contamination and Acanthamoeba keratitis in UK.

As recommended, please include the statistical analysis used for this study in the manuscript (as it is included only in the abstract). As a suggestion, including the odds ratio of exposure to tap water in the AK patient vs the corresponding ratio for control group, would be really interesting.

Great job!

6. PLOS authors have the option to publish the peer review history of their article (what does this mean?). If published, this will include your full peer review and any attached files.

Reviewer #1: Yes: Christopher Aaron Rice

Reviewer #2: No

---

## [Author Response · Author response to Decision Letter 0]

15 Jan 2020

Major changes

• As your study was designed to search for Acanthamoeba and then record the presence of other FLA I would switch this throughout the text. “To determine whether Acanthamoeba keratitis (AK) patients have higher rates of Acanthamoeba or other free-living amoeba (FLA) colonising domestic sinks than control contact lens (CL) wearers, and whether these isolates are genetically similar to the corneal isolates identified from their CL associated AK.”

Answer: The sentence has been modified as suggested. 

• There is no description of the other FLA identified, can you describe which others were investigated and found? 

o Can this also be broken down within the bar graphs? 

Answer: Classification of FLA were not attempted in this study and this was mentioned as a limitation of the study.

“A limitation of this study is that the genus-level identification of FLA by molecular methods was not performed. FLA such as Hartmannella and Vahlkampfiid amoebae [4, 5], although rare compared to Acanthamoeba, are also causative agents of keratitis. In addition, the significantly higher FLA positive samples in the patient cohort suggests that their identification in the genus-level will help to better understand the risk of keratitis associated with domestic tap water. Further studies regarding the pathogenicity of FLA and Acanthamoeba will also require assessing the source of transmission as well as the severity of AK”

• The quality of all of the figures need to be improved (use graph pad prism or another software for better images). 

Answer: Figures have been reformatted as suggested

Figure 1. 

• Do these graphs describe the full picture? Although the graphs tell the reader total numbers it doesn’t break down the exact number of samples obtained from the spout from the kitchen of an AK wearer or the drain from the bathroom of a CL control as examples. A reorganization and coloration to be more descriptive may be useful. 

Answer: Figures have been reformatted to keep numbers of samples. 

Figure 2. 

• As you are specifically identifying different FLA, this graph could be a stacked bar graph with different colors representing the different genus of FLA found for both groups and areas swabbed. 

Answer: Figures have been reformatted as suggested

 As mentioned above genus level identification for FLA was not attempted. 

Minor Changes

• Lines 19-21, 37-41, 79, 113 – switch Acanthamoeba and FLA order.

Answer: The order has been corrected as suggested throughout the manuscript. 

• Don’t you think using E. coli and 32°C selects for Acanthamoeba over other FLA? 

Answer: FLA could be thermophilic (Hartmannella vermiformis and Naegleria) as indicated in this paper https://www.ncbi.nlm.nih.gov/pmc/articles/PMC3469187/. Data of the current study showed that rate of isolation of FLA was higher than that of Acanthamoeba. Therefore, we believed that the incubation condition does not select one over the other. 

• There was no discussion to where the patients were from within the UK (distribution, were these all local to a particular area? – This data should be easily accessible since collected during the questionnaire). Can you make a conclusion about geographic location, potential differences in water storage methods, sources that the water was from or treatment methods? You may find that there was a particular area that could have an increased risk or contamination. This could be useful for water regulatory and treatment of such bodies of water. 

Answer: Yes, we agree that this information would be useful in many ways. However, we were able to collect this information from less than 1/4 of case and controls. Therefore, this analysis has not been added in the manuscript. 

• Why were only 4 patients out of 129 compared for Ac coxA? Please discuss.

Answer: Due to time frame of the project, resources and funding availability, we have selected samples from only four patients for genetic characterization. These four patients appeared to have striking association with detection of Acanthamoeba in water sample from their homes. Our aim was to show relatedness between pathogenic and environmental isolates. We have clarified this in manuscript as 

Four AK patients were selected to the study genetic relatedness between pathogenic and environmental isolates on the basis of association between case and detection of Acanthamoeba in water samples from their homes. Partial sequence of the mitochondrial cytochrome oxidase gene (coxA 1/2) was obtained from Acanthamoeba isolates of four different keratitis patients and those sequences were compared with isolates from the patient’s home.

Lines 24-26 – “The samples were cultured by inoculating onto a non-nutrient agar plate seeded with Escherichia coli and the plates were incubated at 32˚C and examined for amoebae by microscopy for up to 2 weeks.” Could read as “The samples were cultured by inoculating onto a non-nutrient agar plate seeded with Escherichia coli, incubated at 32˚C and examined for amoebae by microscopy for up to 2 weeks.”

Answer: The sentence has been corrected as suggested 

• Lines 30-36 – Please add the %’s for spouts, drains, overflow in kitchens and bathrooms.

Answer: Percentage for sampling sites have been added 

“Spouts (kitchen 6.7%, bathroom 11%) had the lowest rate of Acanthamoeba isolation compared to drains (kitchen 18.2%, bathroom 27.9%) and overflow (kitchen 39.1%, bathroom 25.9%) either in kitchens or bathrooms (p<0.05). There was no statistically significant difference between the prevalence of Acanthamoeba in all three sample sites in kitchens (16.9%) compared to all three sample sites in bathrooms (21.5%) and no association for Acanthamoeba prevalence between AK patients and CL controls.”

• Line 36 – Was this the kitchen or bathroom sink? 

Answer: It was from both the kitchen and the bathroom. The sentence has been modified as 

“All four corneal isolates had the same coxA sequence as at least one domestic water isolate from the patients’ sink of the kitchen and the bathroom.” 

• Line 46 – Change “potentially” to opportunistically. Where these the different types of FLA identified within this study? Please make reference to which ones?

Answer: This has been changed as suggested.

• Line 71 – Reference to ref 11 not needed here since it is referenced at the end of the sentence.

Answer: The reference has been deleted. 

• Line 95-96 – If 23x repeat samples were taken from 129 patients from 6 locations within the home there would be 17,802 samples? I would remove this sentence. Or please show the succession of each patient over the 23 month period as a supplemental figure. 

Answer: The sentence has been rewritten to avoid this confusion. 

“A total of 23 repeat samples at least one month apart were collected from patient’s kitchen and bathroom” 

• Line 114 – 513 total samples were obtained from AK patients = 6.6 samples per patient? 189 samples from 40 CL controls = 4.7 samples per patient? 

Answer: As mentioned in methods, each patient was provided with six sample collecting pack. However, some of them had not returned all six tubes with sample plus we collected 23 repeat samples from AK patents. This is the reason; the sample numbers are not exact multiple of patient number. To avoid any confusion, the sentence has been modified as

“A total of 513 samples from 77 AK patients and 189 samples from 40 CL controls were retrieved and examined in this study.”

Furthermore, the distribution of sample in figure 1 shows that proportion of samples collected from different sites are different. 

Figure 1. 

• Please add in total numbers for each of the described %’s within the pie chart. 

Answer: This has been updated in figure 1. 

• Please add discussion about why the 1.17% (6 cases) from the Cloakroom of AK patients was removed. 

Answer: This has been added as 

“Samples collected from the cloakrooms of AK patients were excluded from the current analysis because the number of samples were small (1.7%) and there were no cloakrooms samples from CL controls group.” 

• Line 121 – (The Acanthamoeba comparison should be Figure 2 not “Fig 1”. Also spell out figure to keep consistent throughout the MS.

Answer: Corrections made as suggested. 

• Line 123-124 – Do both groups (AK patients and CL control) FLA total include the Acanthamoeba data as well? If so, shouldn’t this be removed/normalized? 

Answer: In the current analysis, FLA group includes Acanthamoeba or FLA. The rate of presence of Acanthamoeba and FLA was analysed separately for each sampled area. Therefore, there were no duplicate isolates in the analysis. To clarify this, following modifications have been added 

“Samples identified with Acanthamoeba were classified as Acanthamoeba positive and those with Acanthamoeba or FLA were classified as FLA in the current analysis.” 

Figure 2. 

• Please shorten the significance bar for the ** between the FLA found in patients and controls. 

Answer: Figure has been reformatted as suggested 

• Please keep consistency between titles of groups for all figures, “AK patients” and “CL controls”. 

Answer: Corrections have been made throughout 

• Line 131-137 – Please include the * for level of significance.

Answer: Correction has been made 

• Line 133-137 – Please spell out figure to keep consistency throughout the MS.

Answer: Corrections have been made throughout

• Line 142 – Please Italicize Acanthamoeba. 

Answer: Corrections have been made throughout

Figure 3&4. 

• Please keep consistency between titles of groups for all figures, “AK patients” and “CL controls”.

Answer: Corrections have been made throughout

• Please shorten the significance bar and center the * for kitchen drain and bathroom overflow. 

Answer: Figure has been reformatted as suggested

Table 1. 

• Please centers align and middle aligns text into the center of all boxes of the table.

Answer: Table has been formatted as suggested.

• References font changes. 

Answer: Font has been changed to match with manuscript’s font. 

Reviewer # 1 

Line 138-139 – Spouts have the fastest flowing and largest volume of water flowing through it with thousands of gallons per month, this may be the reason why you do not see as much bacteria/Acanthamoeba/FLA in that particular area for Fig 3. Should repeat samples be taken from the drain or overflows from these areas for future studies? Please comment/discuss this finding. 

Answer: Discussion has been added (line 191 -194)

In the current study tap spouts tended to have filters on them making it difficult to swab the inside of spouts and this could be the reason for these differences [11]. In addition, spouts have the fastest water flow and large volume of water regularly flows through it, which self-flushes the spouts and may prevent colonisation of FLA.

Do you think that paranoia of patients when sampling they didn’t want to see Acanthamoeba or FLA within their home that they may have stringently cleaned between sampling? 

Answer: While a small number of the first samples changed from positive to negative, a similar amount changed in the other direction in repeat sampling. Therefore, it does not appear that this had an effect.

All figures/graphs are of poor quality please use another software program. 

Answer: figures have been modified for better quality. 

Were attempts to characterize the species/isolate causing the AK in these 4 patients? 

Answer: No, only the coxA gene was sequenced.

Supplemental information is clear for samplers. 

Answer: Participants were given a verbal instruction and provided with written instruction.

Molecular characterization of FLA should become mandatory in future studies. 

Answer: Yes, Authors agree with this and discus it in the manuscript. 

Reviewer #2: 

Overall, it is a very well written case-control study on the association between environmental sources of contamination and Acanthamoeba keratitis in UK.

Answer: Authors like to thank reviewer for taking time to review the manuscript and providing comment on it. 

As recommended, please include the statistical analysis used for this study in the manuscript (as it is included only in the abstract). As a suggestion, including the odds ratio of exposure to tap water in the AK patient vs the corresponding ratio for control group, would be really interesting.

Answer: Thank you for suggesting this. Statistical analysis has been added to in the methods of the manuscript as suggested. Odd ratios have been included in results. 

“Statistical analysis

The Pearson Chi-square test was used to assess whether there was a statistically significant difference in the association between sampling sites. Odds ratios and their 95% confidence intervals (95% CIs) were calculated to measure association between AK cases and detection rate of Acanthamoeba and FLA in AK patients’ and CL controls’ homes.”

Great job!

---

## [Editor Report · Decision Letter 1]

12 Feb 2020

The relationship between environmental sources and the susceptibility of Acanthamoeba keratitis in the United Kingdom

PONE-D-19-26513R1

Dear Dr. Carnt,

We are pleased to inform you that your manuscript has been judged scientifically suitable for publication and will be formally accepted for publication once it complies with all outstanding technical requirements.

With kind regards,

Manuel Garza León

Academic Editor

PLOS ONE
---

## [Editor Report · Acceptance letter]

14 Feb 2020

PONE-D-19-26513R1 

The relationship between environmental sources and the susceptibility of Acanthamoeba keratitis in the United Kingdom 

Dear Dr. Carnt:

I am pleased to inform you that your manuscript has been deemed suitable for publication in PLOS ONE. Congratulations! Your manuscript is now with our production department. 

With kind regards,

on behalf of

Dr. Manuel Garza León 

Academic Editor

PLOS ONE